# Transcriptome Analysis of the Hippocampus in Domestic Laying Hens with Different Fear Responses to the Tonic Immobility Test

**DOI:** 10.3390/ani15131889

**Published:** 2025-06-26

**Authors:** Jingyi Zhang, Min Li, Liying Pan, Ye Wang, Hui Yuan, Zhiwei Zhang, Chaochao Luo, Runxiang Zhang

**Affiliations:** 1College of Animal Science and Technology, Northeast Agricultural University, Harbin 150030, China; zhangjingyi0666@163.com (J.Z.); panliying0428@163.com (L.P.); wy505913@163.com (Y.W.); huiyuan@neau.edu.cn (H.Y.); 2College of Life Sciences, Shihezi University, Shihezi 832003, China; btnyjlm@163.com; 3School of Medicine, Shihezi University, Shihezi 832003, China; zzwneau@163.com

**Keywords:** fear behavior, laying hens, hippocampus, transcriptome

## Abstract

Fear behavior is associated with negative emotional responses, which can adversely affect animal welfare. A comprehensive understanding of the neurobiological mechanisms underlying fearfulness is essential for developing effective strategies to improve welfare. In this study, laying hens were classified into high- and low-fear groups based on tonic immobility tests, and hippocampal histological and transcriptomic characteristics were compared to explore the biological basis of individual differences in fear responses. The results revealed that hens with high-fear levels exhibited a significant reduction in Nissl bodies in hippocampal neurons, increased doublecortin (DCX) expression, and decreased c-Fos (Fos proto-oncogene) expression. Differentially expressed genes were enriched in several pathways related to neural function and immune regulation. This suggests that individuals with high fear may experience impaired neural plasticity and dysregulated stress responses. These findings provide molecular-level insights into the mechanisms underlying fearfulness differences and offer a theoretical foundation for future interventions aimed at experiencing less fear and enhancing welfare in laying hens.

## 1. Introduction

Fearfulness is an important behavioral trait in poultry, reflecting an individual’s capacity to respond to potential threats or stimuli. While an appropriate level of fearfulness helps animals avoid danger and improve survival, an excessive or prolonged fear response may adversely affect their health, welfare, and production performance [1,2]. Poultry are usually highly sensitive to external stimuli, and an intense fear response may induce harmful behaviors such as feather pecking, escape attempts, and convulsions [3,4]. In severe cases, fearfulness can lead to a decline in egg production or even trigger panic and trampling, resulting in significant economic losses in commercial farming system [5]. The formation of fear responses is influenced by multiple factors, including genetic background, rearing environment, and early-life experiences [6,7,8].

However, even under identical rearing conditions, individual laying hens still exhibit marked differences in fear responses. Whether these behavioral discrepancies are associated with specific patterns of gene expression in brain tissues remains unclear and warrants further investigation. To accurately classify individual fear states, researchers typically rely on standardized behavioral tests.

Among these, tonic immobility (TI) is a commonly used behavioral test for evaluating the fear response in animals. TI is a typical passive defense response characterized by short-term muscle stiffness and immobility after an animal is stimulated and flipped into a supine position [8]. The duration of TI is closely associated with an individual’s level of fearfulness, with longer durations generally indicating a higher degree of fear [9]. This method is based on the instinctive response of hens using a feigned death strategy to evade predators. It offers several advantages, including ease of operation, a high success rate of induction, and good repeatability. Therefore, TI has been widely applied in studies of fearfulness in poultry to effectively quantify individual fear states [10,11].

In addition to behavioral assessments, increasing attention has been paid to the neurobiological mechanisms underlying fear responses. The hippocampus is an important brain region closely associated with learning, memory, and emotional regulation [12,13], and it plays a broad role in the recognition of environmental stimuli and the regulation of stress responses [13]. Numerous studies have indicated that the hippocampus plays a critical role in the formation and expression of fear memory and that its neural activity and plasticity are closely associated with an individual’s fear behaviors [14,15]. Although the mechanisms underlying hippocampal function in fear regulation have been well characterized in mammals, the relationship between the hippocampus and fear behavior in poultry remains poorly understood.

The present study was conducted to shed some light as to how individual differences affect the molecular regulatory basis of fear behavior responses in laying hens. Therefore, this study classified laying hens into high- and low-fearfulness groups based on the duration of tonic immobility and used these individuals as experimental subjects. Given the key role of the hippocampus in emotion regulation and stress adaptation, we hypothesized that laying hens with high fear responses may have neural structural changes in their hippocampal tissue, accompanied by differences in the expression of genes related to neural plasticity and emotion regulation. These changes may reflect the potential neurobiological mechanisms of individual differences in fear behavior.

## 2. Materials and Methods

### 2.1. Animal Feeding and Housing Conditions

A total of 80 75-week-old Lindian chickens, a local breed from Harbin, China, were used in this study. These chickens are predominantly distributed in Lindian County, Heilongjiang Province, and are well-adapted to the region’s cold northern climate. As a dual-purpose breed, they exhibit both meat and egg production traits [16]. Lindian chickens are noted for their excellent cold resistance, high vitality, and adaptability to roughage [17]. They are medium-sized birds with long shanks, and some individuals possess leg feathers. Their skin is typically white with dense plumage, and their feather coloration varies among individuals, including dark yellow, pale yellow, and black [17].

For individual identification, each chicken was fitted with a numbered ankle bracelet on its left leg. The experiments were conducted in an artificial climate chamber, with the laying hens individually housed in conventional battery cages measuring 48 cm × 42 cm × 35 cm (length × width × height). The lighting schedule provided 14 h of continuous light per day (from 5:00 AM to 7:00 PM) at an intensity of 15–20 lux. The ambient temperature in the barn was maintained at 18–21 °C, with relative humidity levels between 50% and 70%. The hens were fed a commercial layer diet (Datang Minsheng, Harbin, China), and each cage was equipped with individual nipple drinkers to ensure ad libitum access to food and water throughout the experimental period (Appendix A).

### 2.2. Tonic Immobility Test

A total of 80 laying hens at 76 weeks of age were tested for tonic immobility reference using Salzen’s experimental method [18]. Briefly, the hens were removed from the cage and placed belly up in a U-shaped trough. An experienced experimenter then covered the hen’s head with one hand and gently pressed on its breast with the other hand for 15 s. After the chicken stopped struggling, the experimenter slowly removed the hand from the chicken’s breast and walked away until it was no longer in the chicken’s field of view. The time it took for the chickens to return to normal was recorded, and if this duration was less than 15 s, the induction procedure was repeated. If the induction was repeated more than three times, the chickens were considered to have failed the induction. The maximum duration of this test was not more than 15 min, and at the end of the test, the chickens were returned to their original cages and kept until the end of the test.

Among the 80 hens tested, 75 successfully entered tonic immobility, while 5 hens failed to be induced after three attempts and were excluded from further analysis.

### 2.3. Sample Collection

Based on the duration of tonic immobility, nine hens with the longest duration and nine hens with the shortest duration were assigned to the high-fear (TH) and low-fear (TL) groups, respectively [9]. The hens were humanely euthanized via manual cervical dislocation performed by trained personnel, following the guidelines established by the American Veterinary Medical Association (AVMA) and the Canadian Poultry Euthanasia Guidelines. This method rapidly separates the skull from the first cervical vertebra, resulting in immediate unconsciousness and irreversible death. It is recognized as a humane and effective technique for laying hens. The loss of consciousness in each bird was confirmed by the absence of brainstem reflexes (e.g., corneal and palpebral reflexes), and death was subsequently verified. After euthanasia, the left brains of three randomly selected hens from each group were collected and immersed in 4% paraformaldehyde for 10 h, followed by further fixation for histological examination. Additionally, hippocampal tissues from another six selected laying hens were collected into tubes, rapidly frozen in liquid nitrogen, and stored at −80 °C for subsequent transcriptome sequencing and quantitative real-time PCR (qRT-PCR) analysis.

### 2.4. Hematoxylin and Eosin Stain

The fixed left hemispheres of the chicken brains were embedded in paraffin and sectioned at a thickness of 5 μm. Sections were immersed in various concentration gradients of xylene and anhydrous ethanol, followed by staining with hematoxylin and eosin. After transparency treatment, they were embedded in neutral gum and observed and photographed using an orthogonal white light photomicrographic microscope (Nikon Eclipse Ci-L, Nikon, Tokyo, Japan). During microscopic examination, the hippocampal regions were identified within the stained sections of the left hemisphere based on anatomical landmarks. The histomorphometric structure of the hippocampus was then evaluated under high magnification (40×) using CaseViewer 2.4 (3DHISTECH Ltd., Budapest, Hungary).

### 2.5. Nissl Stain

The sections underwent deparaffinization using a dewaxing solution to eliminate waxes, followed by dehydration with immersion in various ethanol gradients. Subsequently, staining was performed using methylamine blue stain, and differentiation was achieved with glacial acetic acid. The processed sections were then subjected to oven drying and transparency treatment. Finally, the sections were sealed under neutral gum and observed and photographed under 40× magnification using an orthogonal white light photomicroscope (Nikon Eclipse Ci-L, Nikon, Tokyo, Japan). The photographs were taken, and the Nissl body area was quantified using Fiji Image J 1.54p software (National Institutes of Health, Bethesda, MD, USA).

### 2.6. RNA Extraction, Purification, and Library Construction

The concentration and purity of total RNA was detected using the Thermo Scientific Nano Drop 2000. The A260/A280 ratio was between 1.8 and 2.0 to ensure RNA purity. In addition, the A260/A230 ratio was assessed to ensure that there is no contamination. RNA integrity assays were performed using RNA-specific agarose electrophoresis with RNA 6000 Nano kit 5067-1511. A total of ≥1 µg of total RNA was selected and enriched for with poly A-tailed mRNA using oligo (dT) magnetic beads and the Next Ultra II RNA Library Prep Kit for Illumina (New England Biolabs, Inc., Ipswich, MA, USA). This was followed by random interruption of the mRNAs by ionic interruption using divalent cations. The fragmented mRNAs were used as templates and random oligonucleotides were used as primers to synthesize cDNAs. cDNAs were purified from double-stranded cDNAs. This was followed by double-end repair and the introduction of “A” bases at the 3′ end. Then, cDNAs were ligated into sequencing junctions. The cDNAs were screened for cDNAs of 400–500 bp using AMPure XP beads and amplified by PCR, and the PCR products were purified again using AMPure XP beads to obtain the final library. The libraries were analyzed using an Agilent 2100 Bioanalyzer (Agilent Technologies, Palo Alto, CA, USA) and Agilent High Sensitivity DNA Kit (Agilent, 5067-4626). The total library concentration was detected using PicoGreen, and the effective library concentration was quantified using Scientific StepOnePlus Real-Time PCR Systems (Thermo Fisher Scientific, Shanghai, China). Multiplexed DNA libraries were homogenized and mixed in equal volumes. The mixed libraries were gradually diluted and quantified. Then, the libraries were sequenced in PE150 mode on an Illumina sequencer.

### 2.7. Bioinformatic Analysis

The original offline data (raw data) are screened using the personalbio Cloud Platform to separate high-quality sequences (clean data). Sequence alignment against the chicken (*Gallus gallus*) genome was then performed to generate the mapped data. Expression levels for each gene were then calculated, followed by differential expression analyses based on gene expression levels between the high- and low-fear groups. Finally, the Gene Ontology (GO) and Kyoto Encyclopedia of Genes and Genomes (KEGG) databases were queried using the DAVID online tool to perform functional annotation analysis and enrichment analysis on the expression levels of differentially expressed genes.

### 2.8. Quantitative Real-Time Polymerase Chain Reaction (qRT-PCR) Assay for Differentially Expressed Genes

Total RNA was extracted from hippocampal samples of laying hens (n = 9) of the TH and TL groups using the RNAiso Plus kit (Takara, Dalian, China). The concentration of total RNA was determined by spectrophotometric measurements at OD260 using a spectrophotometer (Biochrom, Holliston, MA, USA), and total RNA was assessed based on the OD260/OD280 ratio. RNA purity was also assessed. Then, the FSQ-101 Reverse Transcription Kit (TOYOBO, Osaka, Japan) was used to synthesize cDNA by reverse transcription, and cDNA was synthesized by adding oligo dT primer and reverse transcriptase II reverse transcription as described in the instruction manual. The newly synthesized cDNA was diluted 5-fold and used to prepare the qRT-PCR reaction, and the differential target genes were designed using the NCBI website. (Table 1) The qRT-PCR reaction was performed using the Light Cycler 480 qPCR system (Roche, Rotkreuz, Switzerland). Each 10 µL reaction mixture contained 5 µL of 2X NovoStrart SYBR qPCR SuperMix Plus (Novoprotein, Shanghai, China), 1 µL of diluted cDNA, 0.3 µL of upstream and downstream primers for each gene (10 µM), and 3.4 µL of RNase-free water. The qPCR conditions were as follows: initial heating to 95 °C for 1 min, followed by 40 cycles of 95 °C for 20 s and 60 °C for 1 min. The relative mRNA expression of the target genes was calculated with the 2^−ΔΔCt^ method using β-actin as the internal reference gene.

### 2.9. Hippocampal Tissue Immunohistochemistry

Paraffin sections were deparaffinized and processed for antigen repair, and the sections were placed in PBS (pH 7.4) and washed 3 times with shaking for 5 min each time. To block endogenous peroxidase, sections were placed in 3% hydrogen peroxide solution, incubated for 25 min at room temperature in the dark, and washed 3 times in PBS. For blocking, 3% BSA was added dropwise to cover the tissue, and the sample was sealed for 30 min at room temperature. Add primary antibody dropwise (1:200) and incubate in a wet box overnight. (Table 2) Wash sections, cover tissues dropwise with HRP-conjugated secondary antibody of the species corresponding to the primary antibody, and incubate for 50 min at room temperature. DAB color development was performed. DAB color development solution was added dropwise to control the color development time, and then the sections were rinsed with water. The nuclei were stained, and hematoxylin staining and differentiation were performed. Finally, dehydration and transparency were performed, and the sections were sealed with adhesive and photographed for observation under a white light microscope.

### 2.10. Statistical Analysis

Data analysis was performed with SPSS 23.0. Box plots of tonic immobility (TI) duration were plotted using Graphpad 9. Independent samples *t*-tests were used to test for differences between high- and low-fear laying hens with respect to TI duration, blue staining area ratio for Nissl staining, relative mRNA expression levels of target genes determined using qRT-PCR, and immunohistochemical staining of positive spots. Results are expressed as mean ± SEM, with *p* < 0.05 defined as statistically significant and *p* < 0.01 as highly statistically significant.

## 3. Results

### 3.1. Tonic Immobility Test

Based on the same grouping method as our previous study, a total of 75 hens successfully entered tonic immobility and were included in further analysis (a total of 80 hens were subjected to tonic immobilization experiments, of which five hens failed due to more than three inductions) [19]. The results showed that the duration of tonic immobility in laying hens exhibited obvious individual differences and distinctions. The TI duration was very short in the nine hens from the TL group (mean = 65.56 ± 5.30) but very long in the nine hens from the TH group (mean = 858.67 ± 12.57 s) (*p* < 0.001) (Figure 1). Accordingly, these 18 hens, representing individuals with extreme fear responses, were selected for subsequent histological and transcriptomic analyses.

### 3.2. Histological Observations

#### 3.2.1. Hippocampus Tissue H&E Staining

In Figure 2A, the hippocampal neural tissue of the low-fear group chickens exhibited intact cellular morphology, with clearly arranged, healthy appearing cells. No significant signs of inflammatory infiltration or pathological changes, such as neuronal swelling or necrosis, were observed within the examined area. Figure 2B shows a coronal section of the hippocampal tissue from the high-fear group chickens. Compared to the low-fear group, hippocampal neurons in the high-fear group exhibited noticeable shrinkage, accompanied by the presence of vacuoles surrounding the neurons. No inflammatory cells or neuronal necrosis were observed in these sections. Figure 2C presents a sagittal section of the hippocampal tissue from the low-fear group, revealing abundant neurons without any apparent signs of cell death or damage. In contrast, Figure 2D displays a sagittal section of the hippocampal tissue from the high-fear group. Compared to Figure 2C, neuronal density was significantly reduced, and the staining intensity of neurons appeared lighter.

#### 3.2.2. Hippocampus Tissue Nissl Staining

Nissl staining clearly highlighted the neuronal cells in the hippocampal tissue of laying hens. Observations of stained sections from the low-fear (TL) and high-fear (TH) groups were as follows: In the coronal sections of the hippocampus from the TL group (Figure 3A), extensive and intense Nissl staining was observed, with minimal vacuolation. In contrast, the sections from the TH group (Figure 3B) showed a significantly smaller area of intense Nissl staining, accompanied by pronounced neuronal vacuolation. Statistical analysis of the percentage of Nissl-stained blue area in the hippocampal coronal sections of the TL and TH groups (Figure 3C) revealed that the blue staining percentage was significantly higher in the TL group compared to the TH group (*p* < 0.05). Similarly, the results from sagittal sections of the hippocampus (Figure 3D,E) mirrored those from the coronal sections. The TL group exhibited widespread and intense Nissl staining with fewer vacuoles, while the TH group displayed neuronal shrinkage and reduced areas of intense staining. Figure 3F further confirms that the percentage of Nissl-stained blue area in the TL group was significantly higher than in the TH group (*p* < 0.05).

### 3.3. RNA-Seq Analysis

Table 3 demonstrates the basic statistics of the transcriptome sequencing data of the hippocampal tissue of the two groups of laying hens in this experiment. The statistical results showed that the number of raw sequences of each sample was slightly different, but the overall number was similar. The average number of raw sequences per sample was 42,405,879.33. After further data filtering, the number of high-quality sequences obtained reached 38,351,180.33 on average. The proportion of high-quality sequences to the original sequences for each sample was more than 90%, with an average proportion of about 91%. These data validate the high reliability of the sequencing data. In addition, the Q30 percentages of all samples exceeded 92%, indicating that the percentage of bases with a base identification accuracy of 99.9% or more reached more than 92%. This result further emphasizes the accuracy and high quality of the sequencing data.

### 3.4. Differentially Expressed Gene Analysis

Differential gene expression in the hippocampal tissue of high-fear and low-fear groups of laying hens was analyzed using the DESeq2 algorithm. The analysis identified a total of 365 differentially expressed genes (DEGs), including 277 upregulated genes and 88 downregulated genes (Figure 4A). To further illustrate the distribution of these DEGs, a volcano plot was generated. In Figure 4B, red dots on the right represent upregulated genes (log_2_FoldChange > 1), blue dots on the left represent downregulated genes (log_2_Fold change < −1), and gray dots indicate genes without significant differential expression. The volcano plot effectively highlights the differences in gene expression in the hippocampal tissue between the high-fear and low-fear groups.

The results of the differentially expressed gene (DEG) analysis were converted into a hierarchical clustering map (Figure 4C) for intuitive presentation. A horizontal view of the heatmap clearly shows that the three biological replicates from the high-fear and low-fear groups are tightly clustered within their respective groups, indicating a high degree of similarity among samples within each group. This result validates the consistency of gene expression patterns within the groups. From a vertical perspective, the DEGs are distinctly divided into two clusters, including upregulated and downregulated genes, successfully differentiating the two categories of gene expression.

### 3.5. Enrichment Analysis

To understand the functional differences in the differentially expressed genes (DEGs) between the high- and low-fear groups, we performed Gene Ontology (GO) analysis on 365 DEGs, including 277 upregulated and 88 downregulated genes, using the genome background of ‘*Gallus gallus*’. The results of the analysis are shown in Figure 5B, where the top 20 most significantly enriched GO terms are presented (*p* < 0.05). In the GO classification, these DEGs were categorized into three main domains, including Biological Process (BP), Cellular Component (CC), and Molecular Function (MF), corresponding to 563, 56, and 126 terms, respectively. A detailed analysis of these enriched GO terms revealed that functional differences between the high- and low-fear groups were primarily associated with biological functions and molecular activities related to cell membrane structure, intercellular transport, and transmembrane substance transport. In the “Molecular Function (MF)” category, significantly enriched terms included extracellular matrix structural constituent, transmembrane transporter activity, and inorganic molecular entity transmembrane transporter activity. These findings suggest potential differences in substance transport and extracellular matrix structure between the high- and low-fear groups at the molecular level. In the “Cellular Component (CC)” category, enriched terms included extracellular region, collagen-containing extracellular matrix, and collagen trimer, indicating potential differences in cellular structure and extracellular matrix composition between the two groups. In the “Biological Process (BP)” category, significantly enriched terms included skeletal system development, regulation of heart rate by cardiac conduction, and tissue development. These results suggest potential physiological differences in the TH and TL groups.

Kyoto Encyclopedia of Genes and Genomes (KEGG) pathway analysis was performed, and the top 20 enriched pathways were identified, as shown in Figure 5A. From four core domains—Metabolism, Organismal Systems, Cellular Processes, and Environmental Information Processing—we identified seven significantly enriched pathways (*p* < 0.05). These pathways include retinol metabolism, vitamin B6 metabolism, nicotinate and nicotinamide metabolism, neuroactive ligand–receptor interaction, ECM–receptor interaction, pentose and glucuronate interconversions, and glycosaminoglycan biosynthesis—heparan sulfate/heparin. These pathways play essential roles in neural development and immune regulation.

### 3.6. Validation of Differential Gene Expression Using qRT-PCR

To validate the accuracy of RNA-seq data and precisely screen for differentially expressed genes, genes with an absolute log_2_(FoldChange) > 1.4 and an adjusted *p*-value < 0.05 were selected. These genes were associated with neural cell function and synaptic development. Subsequently, real-time quantitative polymerase chain reaction (qRT-PCR) was employed to measure the mRNA expression levels of these differentially expressed genes in the hippocampal tissues of the high-fear and low-fear groups (Figure 6B). To ensure data accuracy, the qRT-PCR results were compared with the transcriptomic data from RNA-seq. As shown in Figure 6A, the trends observed for both methods were consistent, confirming the reliability of the RNA-seq results obtained in this study.

### 3.7. Immunohistochemistry of Hippocampal Tissue

To further verify the association between fear and the hippocampus, protein expression of c-Fos, DCX, and PCNA in the hippocampus was measured using protein immunohistochemistry in laying hens of TH and TL groups (Figure 7), and the results showed that the ratio of the mean optical density value of immunohistochemically positive staining of c-Fos protein in laying hens from the high-fear group was significantly decreased (*p* < 0.05) (Figure 7C), whereas the ratio of mean optical density value of positive staining of DCX was significantly increased (*p* < 0.05) (Figure 7F). No significant change in PCNA was observed (Figure 7I).

## 4. Discussion

### 4.1. Relationship Between Fear Behavior and Hippocampal Tissue Morphology

Negative emotions such as fear are primarily regulated by the central nervous system, with the hippocampus serving as a key brain region involved in emotional regulation and the stress response [20,21]. Based on this, we hypothesize that the morphological and functional changes observed in the hippocampus of highly fearful (TH) laying hens may be closely associated with their ability to recognize and process fear stimuli.

Although no significant neuronal damage or inflammation was detected in the hippocampal tissue, Nissl staining revealed a notable reduction in the Nissl body area in TH hens. As Nissl bodies are the primary sites of protein synthesis within neurons, a reduction in their area could impair the ability of neurons to synthesize and replenish essential proteins, potentially leading to a decline in neuronal function [22]. This observation suggests that TH hens may experience neuronal damage as a consequence of prolonged stress. A study by Ooigawa et al. investigating Nissl staining in the hippocampal and other brain tissues of rats following traumatic brain injury found that hippocampal Nissl bodies are particularly fragile and susceptible to damage under such conditions [23]. Damage to these structures can result in a range of neurological symptoms, including cognitive impairment, motor dysfunction, and sensory abnormalities. Drawing from these findings, we speculate that the prolonged high fear sensitivity exhibited by TH hens may further exacerbate neuronal damage, compounding the adverse effects on hippocampal function.

### 4.2. Effects of Fear Behavior on Hippocampal Gene Expression

Using RNA-seq technology, we analyzed gene expression differences in the hippocampal tissue of chickens with high- and low-fear behaviors. The results showed that the differentially expressed genes were primarily associated with neuronal development and intercellular signal transduction. These genes were significantly enriched in metabolic pathways including retinol metabolism, vitamin B6 metabolism, nicotinate and nicotinamide metabolism, and the neuroactive ligand–receptor interaction pathway. Retinol metabolism involves the interconversion of retinol, retinal, and retinoic acid, playing critical roles in regulating neural development, visual system function, immune responses, and cellular proliferation and differentiation [24]. Studies have shown that the hippocampal cortex contains the highest concentration of retinoic acid in the brain [25], and retinoic acid effectively modulates neural plasticity in the hippocampus, including neurogenesis and synaptic strength. These effects are mediated through specific retinoic acid receptors (RARs) that are highly expressed in the human hippocampus along with related synthesizing enzymes [26]. In chickens with high fear responses, prolonged exposure to stress may cause neuronal damage in the hippocampus, subsequently affecting the expression of enzymes and receptors involved in retinol metabolism. This may lead to reduced activity of the retinol metabolic pathway. Moreover, as chickens are highly visually sensitive animals, alterations in retinol metabolism may also impair visual system function.

Vitamin B6 plays multiple essential physiological roles and is involved in various biochemical metabolic processes in the body. A deficiency in vitamin B6 can lead to anemia, lipid metabolism disorders, and severe skeletal deformities. It also contributes significantly to brain development and the health of the nervous and immune systems. Insufficient vitamin B6 has been associated with depression, anxiety, neuropathy, cognitive decline, and impaired immune responses [27]. Studies have shown that vitamin B6 deficiency can cause hyperactivity of the noradrenergic system in mice, resulting in cognitive impairments and social deficits [28]. Transcriptomic analysis revealed differences in vitamin B6 metabolism between high- and low-fear chickens. This may be due to chronic stress in high-fear individuals, which disrupts vitamin B6 metabolism, potentially leading to reduced availability and increased susceptibility to negative emotional states. Nicotinate and nicotinamide metabolism is closely linked to cellular signal transduction and plays a longstanding role in neurodevelopment and neuroprotection within the central nervous system [29]. Experimental research by Fricker et al. demonstrated that nicotinamide supports energy production and cellular repair during neurodegeneration, findings that have been validated in CNS-related studies [30]. Another study showed that supplementing mice in a depression-like state with nicotinamide mononucleotide helped regulate nicotinamide metabolism and alleviate depressive symptoms [31]. These findings align with our results. Differences in nicotinamide metabolism between the high- and low-fear chickens suggest that neural damage caused by stress in the high-fear group may have activated nicotinamide-related repair pathways as a compensatory response. The neuroactive ligand–receptor interaction pathway modulates neuronal excitability and synaptic transmission efficiency through ligand–receptor binding, such as neurotransmitters and hormones. Under fear-inducing conditions, this pathway likely contributes to the sensitivity and adaptability of the nervous system in response to external stimuli. The ECM–receptor interaction pathway regulates synaptic connectivity, plasticity, and neural repair by mediating interactions between extracellular matrix components and receptors on neuronal membranes. Prolonged stress or fear may result in structural or functional synaptic damage. Changes in this pathway observed between fear response groups may reflect the nervous system’s attempt to maintain synaptic integrity and promote regeneration under chronic stress conditions.

Among the 365 differentially expressed genes identified, several were associated with neuronal function and synaptic development. CCN3, SYNJ2, and OSR1 were upregulated in the hippocampus of high-fear chickens. CCN3 (cellular communication network factor 3) plays a key role in tissue repair and cell growth. In a study by Luan et al., CCN3 knockout mice showed reduced proliferation of hippocampal neural stem cells and increased neuronal differentiation, highlighting its involvement in neurogenesis and repair [32]. This suggests that the elevated expression of CCN3 in high-fear chickens may reflect an adaptive response aimed at promoting neuronal repair and mitigating stress-induced damage in the hippocampus. SYNJ2 encodes synaptojanin 2, a phosphoinositide phosphatase critical for synaptic vesicle recycling and intracellular signal transduction. It facilitates synaptic plasticity and contributes to learning and memory processes by maintaining efficient communication between neurons. Dysregulation of SYNJ2 has been implicated in several neurodevelopmental and neurodegenerative disorders, including Alzheimer’s disease, Parkinson’s disease, and autism spectrum disorder [33], conditions often associated with synaptic dysfunction, chronic inflammation, and impaired plasticity. In the current study, higher SYNJ2 expression in the high-fear group may indicate synaptic stress caused by prolonged emotional pressure. This upregulation could support synaptic homeostasis by enhancing vesicle turnover and restoring synaptic function, acting as part of a compensatory mechanism to protect neural circuits under chronic stress conditions. OSR1 (odd-skipped related transcription factor 1), another gene upregulated in high-fear chickens, is a transcription factor involved in organogenesis, neurodevelopment, and the differentiation and survival of neurons. It also contributes to stress adaptation by modulating relevant signaling pathways in response to environmental and emotional stimuli [34]. Its increased expression may reflect an intrinsic neuroprotective response to emotional challenges in the high-fear phenotype. In contrast to the above genes, PIK3R6 was significantly downregulated in the hippocampus of high-fear chickens. This gene encodes a regulatory subunit of PI3K (phosphoinositide 3-kinase), which modulates PI3K complex activity and is involved in learning, memory, neural plasticity, and the regulation of neuroinflammation [35]. The reduced expression of PIK3R6 may suggest impaired activation of the PI3K pathway in high-fear individuals, potentially leading to decreased neuroadaptive capacity under persistent fear or stress exposure. This disruption could compromise the brain’s ability to respond effectively to chronic emotional stimuli, increasing vulnerability to neuronal dysfunction.

To compare the neural development of chickens with different fear levels, we performed immunohistochemical analysis targeting proteins closely related to neurogenesis. One of the key markers, doublecortin (DCX), is widely used to identify immature neurons undergoing migration and early-stage development. However, under certain stress conditions, DCX expression may not solely reflect neurogenesis—it can also indicate adaptive neural responses to environmental challenges [36]. In this study, high-fear chickens exhibited a significant increase in DCX-positive cells in the hippocampus. Given that all birds were raised under identical environmental conditions, we speculate that the stress experienced by high-fear individuals may arise from social factors, such as aggression from pen-mates, underlying health issues, or, more plausibly, intrinsic differences in stress sensitivity between individuals. These birds may be more reactive to mild but chronic environmental stressors. Elevated DCX expression might represent a compensatory mechanism, supporting neurogenesis and integration of new neurons to help maintain neural function and adaptability under prolonged psychological pressure. This interpretation is supported by findings from Maheu et al., who reported that increased DCX levels may be associated with the extinction of fear responses [37]. Based on this, we further hypothesize that the upregulation of DCX in high-fear chickens could reflect an effort by the nervous system to promote recovery and resilience in the face of persistent fear-related stress. We also examined c-Fos, a protein encoded by one of the immediate early genes (IEGs), which is rapidly and transiently expressed in neurons in response to external stimuli. c-Fos is involved in synaptic plasticity and memory formation [38]. Emotional stimuli, such as fear, can induce DNA double-strand breaks in hippocampal neurons—events thought to reflect both increased transcriptional activity and heightened synaptic remodeling. c-Fos has been shown to mitigate such damage by upregulating neurotransmitter, neuropeptide, and repair-related genes, contributing to neuronal stability through the activation of DNA repair pathways [39]. Based on these roles, we expected increased c-Fos expression in high-fear chickens to support neuronal repair. However, contrary to expectations, c-Fos protein levels in the hippocampus were significantly lower in the high-fear group. Previous studies have shown that c-Fos expression is highly time dependent. Under acute stress, its levels rapidly increase as neurons become activated, facilitating synaptic adaptation. However, during chronic or repeated stress, c-Fos expression tends to decline, potentially as a protective strategy to prevent neural overexcitation, or due to HPA axis-mediated negative feedback and neuronal dysfunction [40,41]. Although all chickens in this study shared the same housing environment, individual sensitivity to environmental stimuli likely varied. High-fear chickens may have been more reactive to subtle environmental cues, maintaining a heightened state of alertness that, over time, led to suppression of c-Fos expression—possibly reflecting a state of neural inhibition or chronic adaptive downregulation. In contrast, low-fear chickens, being less sensitive to the environment, may have shown a stronger neural response specifically to the tonic immobility test, which presented a more intense, acute stressor. This could explain the observed upregulation of c-Fos in the low-fear group relative to their high-fear counterparts.

## 5. Conclusions

This study demonstrates a strong association between tonic immobility duration and fear response in laying hens. Birds with longer immobility times exhibited marked changes in multiple fear-related indicators. These include reduced Nissl body area in the hippocampus; altered neural regulatory pathways such as retinol, vitamin B6, and nicotinamide metabolism; upregulation of neuroregeneration-related genes such as DCX, CCN3, and SYNJ2; and downregulation of c-Fos. Collectively, these findings indicate that laying hens with different tonic immobility durations show distinct differences in neural morphology and molecular markers associated with fear responses.

## Figures and Tables

**Figure 1 animals-15-01889-f001:**
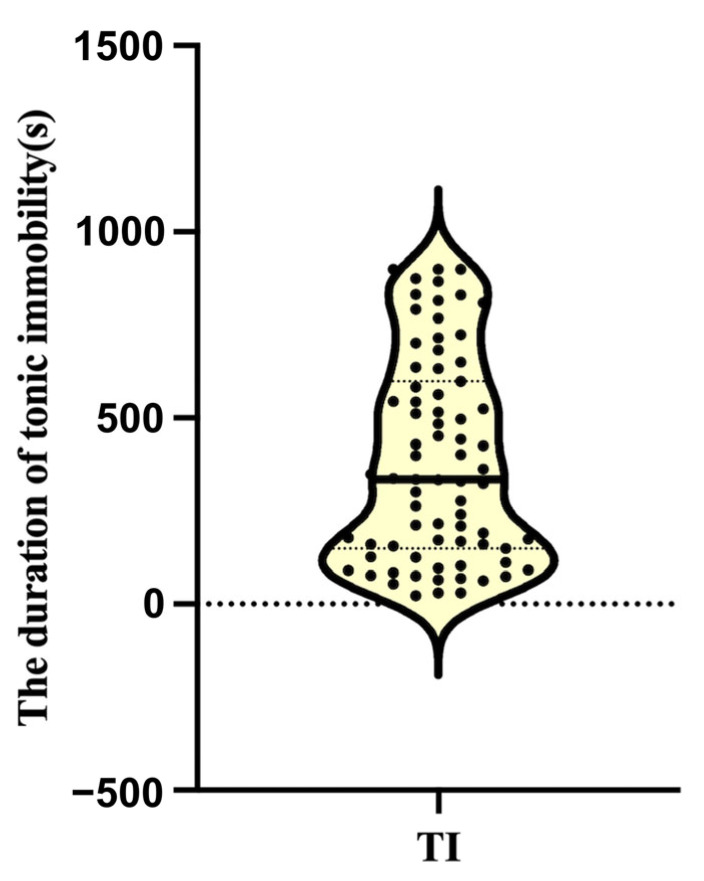
Duration of tonic immobility in laying hens. The solid line represents the median, and the dotted lines represent the quartiles (Q25, Q75).

**Figure 2 animals-15-01889-f002:**
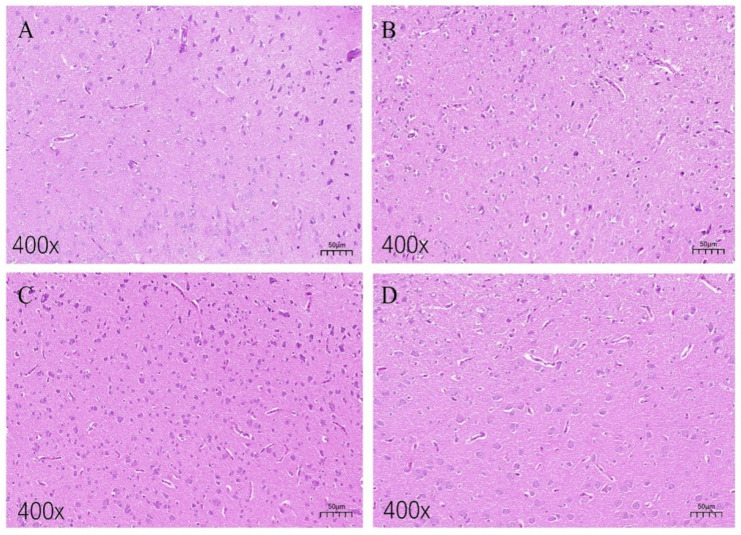
H&E staining images of hippocampal tissue sections. (**A**,**B**) Coronal sections of hippocampal tissues of laying hens in the TL group and TH group, respectively; (**C**,**D**) sagittal sections of hippocampal tissues of laying hens in the TL and TH groups, respectively.

**Figure 3 animals-15-01889-f003:**
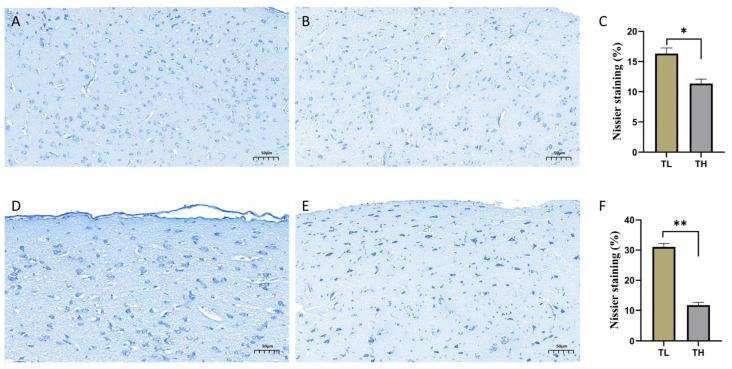
Nissl staining images. (**A**) Coronal section results of the hippocampus in the TL group. (**B**) Coronal section results of the hippocampus in the TH group. (**C**) Bar graph of the percentage of blue-stained Nissl body area in coronal sections of the TL and TH groups. (**D**) Sagittal section results of the hippocampus in the TL group. (**E**) Sagittal section results of the hippocampus in the TH group. (**F**) Differences in the percentage of blue-stained Nissl body area between the TL and TH groups. * *p* < 0.05 and ** *p* < 0.01.

**Figure 4 animals-15-01889-f004:**
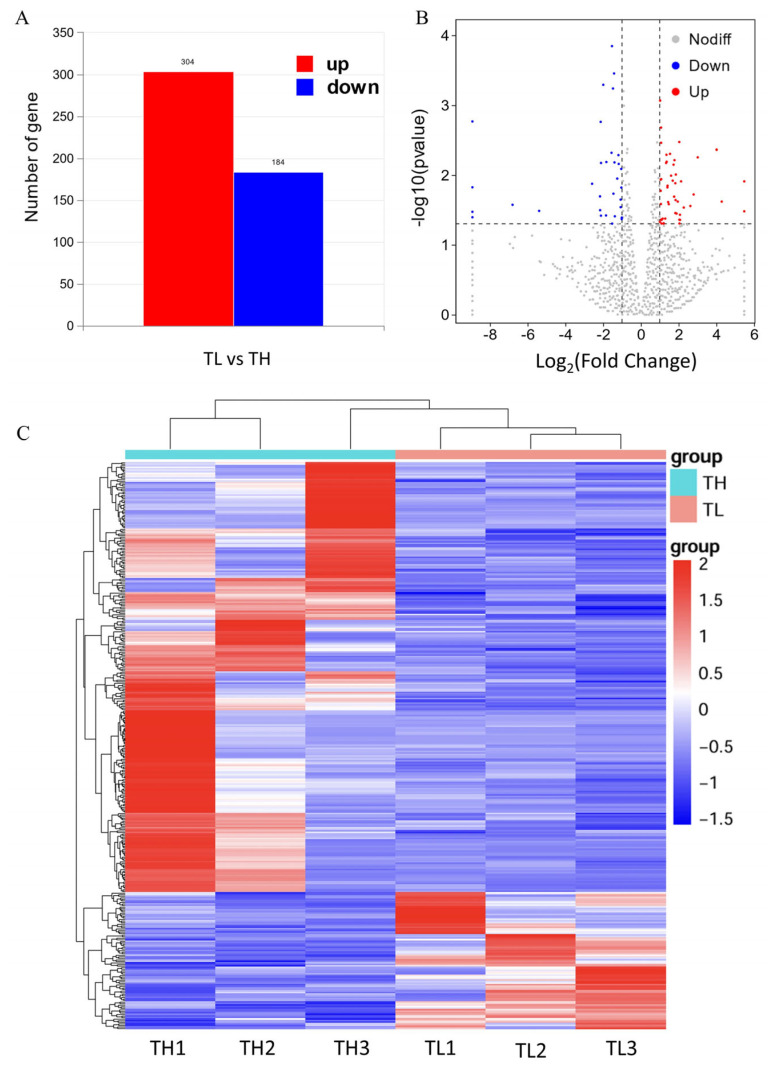
Schematic diagram of differentially expressed genes. (**A**) Differential gene expression bar graph. (**B**) Differential gene expression volcano graph. The horizontal dotted line represents *p* < 0.05, and the area between the two vertical dashed lines represents |log_2_FoldChange| <1. (**C**) Cluster analysis of gene expression in the hippocampus for transcriptome sequencing. Red represents up-regulation, and blue represents down-regulation.

**Figure 5 animals-15-01889-f005:**
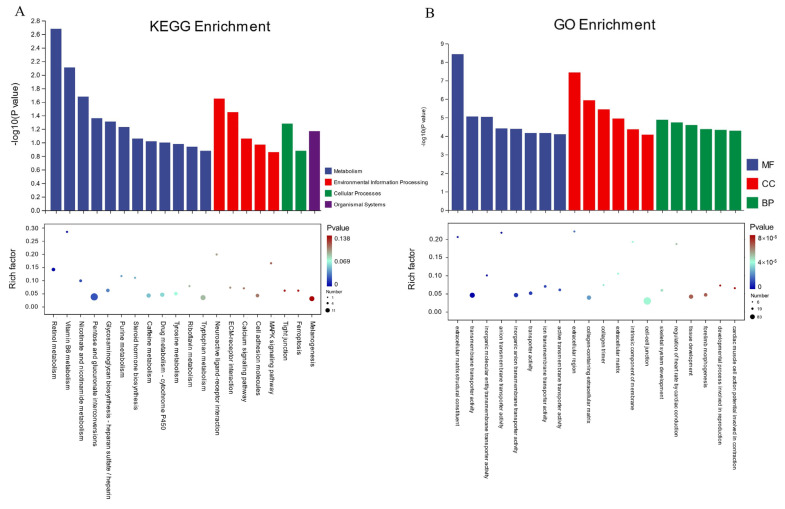
Enrichment analysis results. (**A**) KEGG enrichment analysis. (**B**) GO enrichment analysis.

**Figure 6 animals-15-01889-f006:**
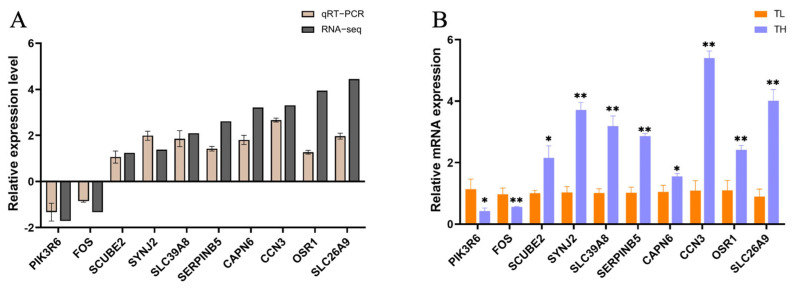
Validation of differentially expressed genes from transcriptome sequencing. (**A**) qRT-PCR verification of differential gene expression trends. The horizontal axis in the figure is the gene name, and the vertical axis is the FoldChange difference in gene expression. (**B**) mRNA expression levels of differentially expressed genes. The horizontal axis in the figure represents the gene name, and the vertical axis represents the relative expression of mRNA. “*” indicates a significant difference with *p* < 0.05; “**” indicates an extremely significant difference with *p* < 0.01.

**Figure 7 animals-15-01889-f007:**
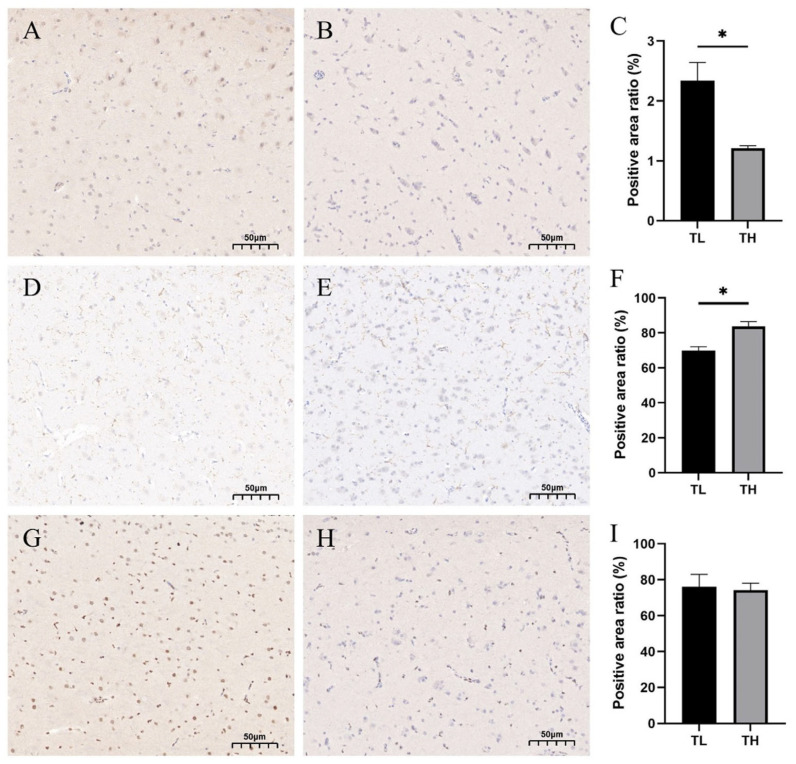
Immunohistochemistry results chart. (**A**) Coronal section of the hippocampus in the TL group. (**B**) Coronal section of the hippocampus in the TH group. (**C**) Difference in c-Fos protein-positive points between the TL and TH groups. (**D**) Sagittal section of the hippocampus in the TL group. (**E**) Sagittal section of the hippocampus in the TH group. (**F**) Difference in DCX-positive points between the TL and TH groups. (**G**) Sagittal section of the hippocampus in the TL group. (**H**) Sagittal section of the hippocampus in the TH group. (**I**) Difference in PCNA-positive points between the TL and TH groups. * *p* < 0.05.

**Table 1 animals-15-01889-t001:** Hippocampal qRT-PCR gene-specific primers.

Target Genes	Gene ID	Primer Sequences (5′ → 3′)
CCN3	NM_205268.2	F: GTGCTGCGAGAAGTGGGTGTG R: CAAGTGTGGCCTCCTGTCTGTATG
SYNJ2	XM_046915116.1	F: GTCAGAGGCAGAACAGTGAAGATCC R: TGTCCCGTTTCCGAGCAATTTCC
OSR1	XM_046938364.1	F: TGCCCACCTTCCCACTCTTCC R: GGCTGTCCACCTGTCCCAATTTAG
SERPINB5	XM_040663178.2	F: TTAGGGCTGGGAAAGTGGGAGTG R: CCTGTGGCGAAACCTTGCTGAG
SLC26A9	XM_040691172.2	F: CAGTGAGGAAAGAGGAGGGAGGAG R: GGTGGAAGAAGCAGGTGATTCAGAC
PIK3R6	XM_001232430.7	F: GCGAGGTGACTACAGGCATTACATC R: GGGCAGGGTGAGCAGTTTCTTG
SLC39A8	XM_046917109.1	F: AACTTGGCATCGCTTCTGGG R: TGCCTCTGGAATGAGCTGGA
SCUBE2	XM_040673558.2	F: GCTCTGGGTCTGGATCACCT R: CTTCCCGTTCCTGTGATGCC
CAPN6	XM_040670579.2	F: CTGCAGGGATGTGGAGCAAG R: CACACGCTCTTCTAGGCTGC
FOS	NM_205508.1	F: CGGGGACAGCCTCACCTACTAC R: GGTCGGGACTGGTGGAGATGG
β-actin	NM_205518.1	F: CACCACAGCCGAGAGAGAAAT R: TGACCATCAGGGAGTTCATAGC

**Table 2 animals-15-01889-t002:** Immunohistochemistry target antibodies.

Antibody	Origin	Dilution Factor
DCX (Servicebio, Wuhan, China)	Rat	1:200
PCNA (Servicebio, Wuhan, China)	Rat	1:500
c-Fos (Servicebio, Wuhan, China)	Rat	1:200

**Table 3 animals-15-01889-t003:** Basic data statistics, collation, and analysis of transcriptome sequencing of the hippocampus in two groups of laying hens.

Sample	Original Sequence	High-Quality Sequences	High-Quality Sequences (%)	Q30 Quantity	Q20 (%)	Q30 (%)
TH1	43,762,124	39,754,198	90.84	6,072,260,563	97	92.5
TH2	41,071,746	37,405,486	91.07	5,722,248,385	97.16	92.88
TH3	45,583,604	41,816,976	91.73	6,300,372,204	96.85	92.14
TL1	41,488,308	37,034,350	89.26	5,764,642,478	97.06	92.63
TL2	41,732,504	37,203,070	89.14	5,810,482,311	97.12	92.82
TL3	40,796,990	36,893,002	90.43	5,676,070,386	97.11	92.75

## Data Availability

All data generated or analyzed during this study are included in this published article. Additional data related to this study are available from the corresponding author upon reasonable request.

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
