# Peer review of "Transcriptome Analysis of the Hippocampus in Domestic Laying Hens with Different Fear Responses to the Tonic Immobility Test"

_animals, 2025, doi:10.3390/ani15131889_

Round 1

Reviewer 1 Report

Comments and Suggestions for Authors

The manuscript presents an interesting approach to studying stress responses in chickens by using a behavioural test and comparing it with gene expression and histopathology. The authors used a local chicken breed for this research. This is additional research from another project (doi: 10.1016/j.psj.2024.103816). 

Please find below some comments. 

L2- The title should specify the breed of laying hen

L95- Consider including the hypothesis of the study

L 127- Include how many were considered failed trials.

L 131- Please elaborate more on why only 18 out of 80 were included, or justify why this is a valid sample size. 

L 132- Has this type of classification been validated? Include references.

L 133- Describe the euthanasia method

L 149- Did a pathologist read the slides?

L  234- Consider moving this up to the methodology sections. Additionally, cite the other publication in the methodology (20).

L 237-238 Include the unit of measurement

L 241- Figure 1 is not referenced within the text.

L 393- Please describe if using a local breed could be a limitation or potential differences with commercial breeds. 

Author Response

We sincerely thank the editor and all reviewers for their valuable feedback that we have used to improve the quality of our manuscript. The reviewer comments are laid out below in italicized font and specific concerns have been numbered.

Comments 1: L2- The title should specify the breed of laying hen

Response 1: We appreciate the reviewer’s valuable suggestion. In response, we have modified the manuscript title to “Transcriptome Analysis of the Hippocampus in domestic laying hens with Different Fear Responses under Tonic Immobility Test

Comments 2: L95- Consider including the hypothesis of the study

Response 2: Thank you for your suggestion. As recommended, we have added the hypothesis at the end of the Introduction section. The revised text reads:

“Given the key role of the hippocampus in emotion regulation and stress adaptation, we hypothesized that laying hens with high fear responses may have neural structural changes in their hippocampal tissue, accompanied by differences in the expression of genes related to neural plasticity and emotion regulation. These changes may reflect the potential neurobiological mechanisms of individual differences in fear behavior.”

This addition clearly states our research hypothesis and aligns with the overall objective of the study. We appreciate your constructive feedback.

Comments 3,8: L 127- Include how many were considered failed trials ; L 234- Consider moving this up to the methodology sections. Additionally, cite the other publication in the methodology (20).

Response 3,8: We appreciate the reviewer’s valuable suggestions. In response, we have revised the Methods section to include the number of hens that successfully completed the tonic immobility (TI) test as well as those that failed induction after three attempts. Specifically, we noted that 75 out of 80 hens successfully entered TI, while 5 hens were excluded due to failure after repeated induction attempts. In both parts you mentioned, the number of laying hens that succeeded in the TI test and the number of laying hens that failed to complete the test and were eliminated are strictly marked.

In fact, we have already mentioned this part (about the number of successful and failed chickens) in the conclusion, which also answers your request in question 8 (that this part should be put in the materials and methods). Now I will give a more detailed response to this question here, and I hope my response can answer your question.

Comments 4: L 131- Please elaborate more on why only 18 out of 80 were included, or justify why this is a valid sample size.

Response 4: Thank you for your thoughtful review. We have clarified the selection criteria for these 18 hens in both the Materials and Methods and Results sections. Specifically, we selected the 9 hens with the shortest TI duration and the 9 hens with the longest TI duration to represent the two extreme fear states to ensure a clear contrast between high-fear and low-fear phenotypes. This approach allows for more robust detection of differences in hippocampal morphology and gene expression.

Comments 5: L 132- Has this type of classification been validated? Include references. Thank you for your important question. We will respond in detail from the following aspects:

First, the rationality of TI test as a tool for assessing animal fear level has been supported by a large number of studies. In a review of various methods for assessing fear behavior in animals, Forkman et al. (2007) pointed out that TI is a behavioral tool widely used in poultry research. Its behavioral performance originates from the "freezing" state induced by simulated predation events and has clear evolutionary adaptive significance. The review also mentioned that the duration of TI has good repeatability, and stress treatment will prolong the TI time, while environmental enrichment will shorten the TI time. Therefore, from a methodological point of view, TI has been widely accepted as an indicator for assessing poultry fear and has a physiological basis.

Second, in response to your question about the grouping based on TI in this study, we conducted TI tests on all experimental chickens in the experiment, and selected the 9 chickens before and after TI (the longest and the shortest) as the high fear (TH) group and the low fear (TL) group based on the duration of TI. As confirmed by the above review and a large number of studies, the duration of TI is significantly correlated with the individual's fear level, so this grouping method has a theoretical basis.

If the above methodological basis is still not enough to dispel your doubts, we also hope to provide support from the results level. In this study, laying hens in the high fear group showed higher sensitivity to external stimuli, accompanied by neuronal damage and changes in hippocampal tissue structure and function, and showed significant differences from the low fear group at the molecular level related to neurogenesis (proliferation, migration and differentiation). These differences are not only reflected in behavioral phenotypes, but also in tissue morphology and molecular levels, further verifying the biological validity of our grouping from the perspective of results.

Finally, we would like to add a study previously published by our laboratory (Wang et al., 2024, Poultry Science), which also grouped laying hens into high and low fear based on TI duration, and systematically evaluated the differences between the two groups in immune factors, intestinal inflammatory factors, intestinal barrier gene expression and intestinal flora composition. The results clearly showed that the TI test can be used as an effective fear level assessment tool, and the grouping criteria have good explanatory power. This study is a further extension and deepening of this study, so we have reason to believe that this grouping method is reasonable and effective in both theory and practice.

[1]Forkman B, Boissy A, Meunier-Salaün M C, et al. A critical review of fear tests used on cattle, pigs, sheep, poultry and horses[J]. Physiology & behavior, 2007, 92(3): 340-374.

[2]Wang, Y.; Zhang, J.; Wang, X.; Wang, R.; Zhang, H.; Zhang, R.; Bao, J. The Inflammatory Immunity and Gut Mi-crobiota Are Associated with Fear Response Differences in Laying Hens. Poult Sci 2024, 103, 103816, doi:10.1016/j.psj.2024.103816.

Comments 6: Describe the euthanasia method.

Response 6: Thank you for highlighting this important detail. We have now expanded the Materials and Methods section to include a full description of the euthanasia procedure as follows: “Following completion of the experimental procedures, hens were euthanized by manual cervical dislocation performed by trained personnel in accordance with AVMA and Canadian poultry euthanasia guidelines. This method rapidly dislocates the skull from the first cervical vertebra, causing immediate loss of consciousness and irreversible death, and has been approved as humane and efficient for layer hens. Each bird’s lack of consciousness was verified by the absence of brainstem reflexes (e.g., no corneal or palpebral reflex), followed by confirmation of death.” This addition ensures compliance with current animal welfare standards and enhances the reproducibility and transparency of the experimental protocol.

Comments 7: Did a pathologist read the slides?

Response 7: Thank you for your question. Yes, all histological slides were reviewed and interpreted by qualified pathologists. Specifically, two co-authors of this manuscript, Dr. Chaochao Luo and Dr. Zhiwei Zhang, who are experienced experts in histopathology, independently examined all tissue sections to ensure accuracy and consistency in the morphological evaluations.

Comments 8: L 237-238 Include the unit of measurement.

Response 8: Thank you for your suggestion. We have added the appropriate unit of measurement at the indicated location to improve clarity.

Comments 9: Figure 1 is not referenced within the text.

Response 9: Thank you for pointing this out. We have carefully revised the manuscript to ensure that Figure 1 is now properly referenced within the main text.

Comments 10: L 393- Please describe if using a local breed could be a limitation or potential differences with commercial breeds. 

Response 10: Thank you for raising this important point. In the present study, we did not investigate the differences in fear behavior across different breeds, as it was beyond the scope of our research objectives. However, we acknowledge that breed-related genetic background could potentially influence fear responses.

For readers interested in this aspect, we would like to refer to a doctor study from our laboratory that specifically examined fear behavior differences among chicken breeds. The findings are published and available at the following DOI:

DOI:https://doi.org/10.1093/jas/skac076

Reviewer 2 Report

Comments and Suggestions for Authors

Transcriptome Analysis of the Hippocampus in Laying Hens with Different Fear Responses under Tonic Immobility Test

Specific Comments 

Line 114-115: “The hens were fed a commercial layer diet (Datang Minsheng, Harbin, China), and each cage was equipped with individual nipple drinkers to ensure ad libitum access to food and water throughout the experimental period”.

Comment: “What are the specific ingredients in the commercial diet?”. 

Suggestions: “Can you include information on the diet used?”

Line 233-234: “Same as the results of the previous study, a total of 80 hens were subjected to tonic immobilization experiments, of which 5 hens failed due to more than 3 inductions [20].”.

Comment: “More detail explanation needed with reference to previous study”. 

Suggestions: “For clarity”

Abstract seems lengthy but it overall covers the research work.

Provide citations and references: Line 58, Line 105, Line 114, Line 178-185, Line 300                 

Explain DESeq2 algorithm.

General Comments  

  1. Clarity: The writing and flow of the manuscript seems clear to me.
  2. Methodology: Explain Bioinformatics section in more detail citing the tools used for sequence analysis.
  3. Results: Figure 1 is missing in the paragraph of Tonic immobility test. Make sure you explain Figure 1 well.
  4. Elaboration: The discussion should explore the practical implications of the findings in greater depth, including limitations of the study too.

      Comments on the Quality of English Language

Author Response

We sincerely thank the editor and all reviewers for their valuable feedback that we have used to improve the quality of our manuscript. The reviewer comments are laid out below in italicized font and specific concerns have been numbered.

Thank you for your valuable comment regarding the language quality of the manuscript. In response to your suggestion, we have carefully revised the entire manuscript for English language and grammar. To ensure accuracy and fluency, the revised version has been thoroughly polished by a native English-speaking Ph.D. in the field. We believe that the current version meets the required language standards for publication.

We sincerely appreciate your helpful feedback, which has contributed to improving the quality of our work.

Comments 1: Line 114-115: “The hens were fed a commercial layer diet (Datang Minsheng, Harbin, China), and each cage was equipped with individual nipple drinkers to ensure ad libitum access to food and water throughout the experimental period”. Comment: “What are the specific ingredients in the commercial diet?”. Suggestions: “Can you include information on the diet used?”

Response 1: Thank you for your valuable suggestions. To this end, we have added the detailed composition of the commercial laying hen diet used in this study in the form of a supplementary table in the appendix. If you are interested in the specific diet formula, please refer to the appendix for more information. I hope this addition will meet your expectations. In addition, for easy reference, I have placed the feed composition table here for your convenience.

Feed Composition Table

Ingredients

Content (%)

Nutritional indicat content ors

Estimated

Corn

65.2

Metabolizable energy (ME)

~2755 kcal/kg

Soybean meal

20.8

Crude protein (CP)

18.6 - 19.4%

Vegetable oil

1.1

Crude fat

5.6 - 6.4%

Powdered rock

3.4

Calcium (Ca)

3.55 - 3.75%

Calcium hydrogen phosphate

0.95

Total phosphorus (P)

0.64 - 0.69%

Salt

0.32

Available phosphorus (AP)

0.36 - 0.41%

Fish meal

2.9

Lysine (Lys)

1.02 - 1.08%

Core material

1.05

Methionine (Met)

0.41 - 0.46%

Choline chloride

0.06

Threonine (Thr)

0.69 - 0.74%

Threonine

0.28

Sodium (Na)

0.16%

Methionine

0.32

Vitamin A

11800 - 12200 IU/kg

Lysine

0.18

Vitamin D3

2900 - 3100 IU/kg

98% lysine

0.29

Vitamin E

25 - 50 mg/kg

Vitamin premix

0.05

Comments 2: Line 233-234: “Same as the results of the previous study, a total of 80 hens were subjected to tonic immobilization experiments, of which 5 hens failed due to more than 3 inductions [20].”. Comment: “More detail explanation needed with reference to previous study”. Suggestions: “For clarity”

Response 2: Thank you for your comments. The previous study we cited [20] was conducted by our research group to explore the differences in intestinal flora and inflammatory responses in laying hens with different fear levels based on the tonic immobility (TI) test. Our current study builds on this study and uses a similar classification method based on TI duration to further explore the structural and genetic differences in the hippocampus of hens with high and low fear responses.

When we mention "same as previous research results", we intend to indicate that :

1)our grouping method - based on TI duration and selecting hens with extreme fear levels - is consistent with the previous method;

2) the grouping results follow the previous grouping method to obtain two groups, high fear group and low fear group, and then continue to conduct subsequent experiments. If you are interested, I can provide you with the paper, in which you can see that the paper also divides laying hens into high fear and low fear groups according to the duration of TI (laying hens that still fail to pass the TI test more than 3 times are eliminated), and then conducts subsequent experiments on these two groups.

I apologize for any confusion caused by the previous wording and have revised this part of the manuscript to improve clarity. Then I change it into : “Based on the same grouping method as our previous study”

Please see the revised version for an updated explanation.

Here is the paper I just mentioned. I put it here with the DOI number for your reference.

(Wang, Y.; Zhang, J.; Wang, X.; Wang, R.; Zhang, H.; Zhang, R.; Bao, J. The Inflammatory Immunity and Gut Microbiota Are Associated with Fear Response Differences in Laying Hens. Poult Sci 2024, 103, 103816, doi:10.1016/j.psj.2024.103816.)

Comments 3: Provide citations and references: Line 58, Line 105, Line 114, Line 178-185, Line 300. Explain DESeq2 algorithm.

Response 3: Thank you for your question. Now I will answer the questions you raised about adding references one by one:

1) The references that need to be cited in line 58 are the same as the next sentence, that is, [1][2].

2) The references that need to be cited in line 105 are the same as the next sentence, that is, [16], which I have already cited.

3) Line 114. In order to reduce the stress of the laying hens we purchase due to environmental changes, our breeding environment is set according to the breeding environment of local laying hen farms, so there is no reference.

4)Lines 178-185. The original data quality control part is based on cloud platform data, which I have marked in the text. There are specific methods on the platform, but there are no references. Readers can obtain knowledge in this regard based on the content of the cloud platform. ( https://www.genescloud.cn/login )

5) Line 300. The method of differential gene analysis uses the deseq2 algorithm. This algorithm is the analysis method provided to us by the cloud platform when we perform differential gene analysis. For specific explanations, you can refer to the cloud platform link I provide to you below.

( https://www.genescloud.cn/login )

Reviewer 3 Report

Comments and Suggestions for Authors

The paper explores interesting topic and uses advanced methods to link fear and fearfulness in laying hens to morphological changes and gene expression levels. The authors, however, are not very specific about their aims, and the hypothesis is missing completely.

The origin of TI is antipredatory behaviour, and this aspect is not mentioned in the paper at all. Although often used as a measure of fearfulness, it also has an adaptive value. As a measure of fear, TI is far from ideal, as it has pure repeatability, duration can be affected by other factors (such as noise or movement in the testing room). Authors based their conclusions on only one measure of fear, while there are other behavioural and physiological measures, to which gene expression and immunohistochemical results could be related.

It is unclear, how the comparison of hippocampal tissue sections of TH and TL hens was made, please add more detail.

Authors state that 9 hens were selected for either TH or TL group. How many hens were then used for  H&E staining, Nissl and gene expression?

Conclusions are vague and general.

To really attribute changes in gene expression to stress resulting from chronic fear, a different experimental design would be needed, with several behavioural and physiological measures.

Despite some drawbacks, the study outlines an important link between chronic stress and neuronal damage.

Author Response

We sincerely thank the editor and all reviewers for their valuable feedback that we have used to improve the quality of our manuscript. The reviewer comments are laid out below in italicized font and specific concerns have been numbered.

Comments 1: The origin of TI is antipredatory behaviour, and this aspect is not mentioned in the paper at all. Although often used as a measure of fearfulness, it also has an adaptive value. As a measure of fear, TI is far from ideal, as it has pure repeatability, duration can be affected by other factors (such as noise or movement in the testing room). Authors based their conclusions on only one measure of fear, while there are other behavioural and physiological measures, to which gene expression and immunohistochemical results could be related.

Response 1: Thank you for your valuable comments. We fully agree with your point of view that tonic immobility (TI) originally originated from the anti-predator instinctive reaction of animals. Although TI is not the only or most ideal means to assess fear, it is still one of the most widely used and behaviorally effective classic methods in poultry behavioral research. Below we will respond to your question from the two levels of theoretical basis and experimental verification in combination with existing literature.

First, in the review of various animal fear behavior assessment methods by Forkman et al., TI was identified as the core fear test method in poultry research. The reasons include: TI test has good repeatability, easy standardization, and based on the predator avoidance behavior in its evolutionary background, it has a clear ecological and evolutionary basis. In addition, the duration of TI can be regulated by the external environment and individual experience. For example, stress will prolong the TI time, while environmental enrichment can significantly shorten the TI time, which reflects the correlation between TI time and animal emotions. Therefore, TI has a strong theoretical basis and empirical support as an assessment tool for poultry fear behavior.

Secondly, although the studies of Campbell et al. pointed out that the TI test has certain limitations in distinguishing complex emotional states (such as anxiety) compared with the attention bias test and the open field test, these studies also emphasized that TI is still a feasible, controllable and practical method for evaluating poultry fear behavior. In actual operation, the TI test is particularly suitable for large-scale screening and typing, and its simplicity of operation has unique advantages in actual research.

We also agree that the combination of multiple behavioral indicators can more comprehensively evaluate the fear state of animals, but the design focus of this study is not to comprehensively evaluate the fear level of laying hens, but to further compare the differences in hippocampal tissue morphology, neuronal activity and related gene expression based on the high and low fear groups formed by the TI test. Therefore, we use the TI test as the only behavioral evaluation indicator for the purpose of research focus, rather than ignoring the value of other behavioral methods.

To further support the scientificity of the division of high and low fear groups based on TI duration in this study, we cited a study previously published by the laboratory (Wang et al., 2024, Poultry Science), which also used TI test to group laying hens into fear groups, and systematically compared the differences in multiple indicators such as immune factors, intestinal inflammatory factors, barrier gene expression and intestinal flora composition between high and low fear groups. The results showed that TI grouping had significant differences and biological validity at the molecular physiological level. This study is based on the further extension of this study, from the intestinal level to the central nervous system, further verifying the applicability and effectiveness of TI behavioral typing in neurobiological research.

In summary, although the TI test has certain limitations as a behavioral assessment tool, its wide application in poultry research, good operability and repeatability, and close association with physiological immune indicators and neurobehavioral parameters fully support the scientificity and rationality of our fear level grouping based on TI test. This study is not intended to establish a comprehensive behavioral assessment system, but to conduct in-depth neurobiological mechanism research on specific fear behavior typing based on the previous verification work. Therefore, we believe that the current behavioral testing and grouping methods are scientific choices based on existing evidence and experimental purposes.

[1]Forkman B, Boissy A, Meunier-Salaün M C, et al. A critical review of fear tests used on cattle, pigs, sheep, poultry and horses[J]. Physiology & behavior, 2007, 92(3): 340-374.

[2] Campbell D L M, Dickson E J, Lee C. Application of open field, tonic immobility, and attention bias tests to hens with different ranging patterns[J]. PeerJ, 2019, 7: e8122.

[3]Wang, Y.; Zhang, J.; Wang, X.; Wang, R.; Zhang, H.; Zhang, R.; Bao, J. The Inflammatory Immunity and Gut Mi-crobiota Are Associated with Fear Response Differences in Laying Hens. Poult Sci 2024, 103, 103816, doi:10.1016/j.psj.2024.103816.

Comments 2: It is unclear, how the comparison of hippocampal tissue sections of TH and TL hens was made, please add more detail.

Response 2: Thank you for your comment. We apologize for the lack of clarity regarding the comparison of hippocampal tissue sections between the TH and TL groups. As described in the Materials and Methods section, hematoxylin and eosin (H&E) staining was performed for qualitative histological observation. The analysis focused on identifying general morphological differences, such as tissue integrity, cell density, and potential signs of structural disruption, rather than quantitative measurements. Therefore, no statistical or morphometric quantification was performed on the stained sections. We have revised the manuscript to clarify this point and emphasize that the H&E staining served primarily as a qualitative assessment of hippocampal tissue morphology.

Comments 3: Authors state that 9 hens were selected for either TH or TL group. How many hens were then used for H&E staining, Nissl and gene expression?

Response 3: Thank you for your question. From each group (TH and TL), 3 hens were randomly selected for histological analysis (H&E and Nissl staining), and 6 hens were used for gene expression analyses, including transcriptome sequencing and qRT-PCR. I have already added this additional information in the "sample collection" section.

Comments 4: Conclusions are vague and general.

Response 4: Thank you for your question. We have revised the Conclusion section and specifically modified it to:“This study demonstrates a strong association between tonic immobility duration and fear response in laying hens. Birds with longer immobility times exhibited marked changes in multiple fear-related indicators, including reduced Nissl body area in the hippocampus, altered neural regulatory pathways such as retinol, vitamin B6, and nicotinamide metabolism, upregulation of neuroregeneration-related genes such as DCX, CCN3, and SYNJ2, and downregulation of cFos. Collectively, these findings indicate that laying hens with different tonic immobility durations show distinct differences in neural morphology and molecular markers associated with fear responses.”

Comments 5: To really attribute changes in gene expression to stress resulting from chronic fear, a different experimental design would be needed, with several behavioural and physiological measures.

Response 5: Thank you for your thoughtful suggestion. As mentioned in our response to Comment 1, the primary aim of this study was to assess the reliability of tonic immobility (TI) duration as an indicator of fearfulness in laying hens. While we acknowledge that a more comprehensive experimental design—including multiple behavioral and physiological assessments—would strengthen the conclusions regarding the relationship between chronic fear and gene expression, numerous studies have validated the use of TI duration as a reliable proxy for fear-related responses in poultry. Related references are as follows:

  • Tiemann, I.; Becker, S.; Fournier, J.; Damiran, D.; Büscher, W.; Hillemacher, S. Differences among Domestic Chicken Breeds in Tonic Immobility Responses as a Measure of Fearfulness. PeerJ 2023, 11, e14703, doi:10.7717/peerj.14703.
  • Jones R B. The tonic immobility reaction of the domestic fowl: a review[J]. World's poultry science journal, 1986, 42(1): 82-96.
  • Gallup G G. Tonic immobility as a measure of fear in domestic fowl[J]. Animal Behaviour, 1979.

Nonetheless, we agree with your valuable suggestion and will incorporate multiple indicators of fear and stress responses in future research to build a more robust and multidimensional evaluation system.

Reviewer 4 Report

Comments and Suggestions for Authors

This study investigated the neurohistological characteristics of the brain tissue and hippocampal transcriptomic differences between high- and low-fear response Lindian chickens. The research findings demonstrate that individuals with high fear levels exhibit neuronal damage, along with abnormalities in neurogenesis, synaptic signaling pathways, and stress-related pathways.

The obtained results are presented accurately and sufficiently support the hypotheses proposed by the authors. The study demonstrates a high standard of scientific research, consistent with current findings in this field. The graphical representation of experimental results and overall discussion are well-structured. This paper possesses significant academic value, though several recommendations for improvement should be addressed as follows:

  1. The experimental animals used in this study were Lindian chickens, a dual-purpose breed for both meat and egg production. Unlike commercialized breeds that typically use only female layers, were all the animals in this study females, or were both sexes included? If both sexes were included, what was their ratio? Additionally, should the potential relationship between sex and fear levels also be considered?
  2. In this study, lines 133-138 describe the use of the left hemisphere of the chicken brain for histological observation, while hippocampal tissues were used for transcriptome sequencing and PCR assays. However, lines 140-147 mention eosin staining being performed on hippocampal tissues. Therefore, was the histological observation conducted on the hippocampal tissues or the left brain? Please clarify.
  3. In lines 396-398, the authors present their hypothesis. If this hypothesis is indeed original and based on their own research findings, why is a reference cited? If a citation is necessary, does this imply that the hypothesis was originally proposed by others and you are merely validating it? Please clarify this discrepancy.

Author Response

We sincerely thank the editor and all reviewers for their valuable feedback that we have used to improve the quality of our manuscript. The reviewer comments are laid out below in italicized font and specific concerns have been numbered.

Comments 1: The experimental animals used in this study were Lindian chickens, a dual-purpose breed for both meat and egg production. Unlike commercialized breeds that typically use only female layers, were all the animals in this study females, or were both sexes included? If both sexes were included, what was their ratio? Additionally, should the potential relationship between sex and fear levels also be considered?

Response 1: Thank you for your insightful question. All animals used in this study were female Lindian chickens, as the breed is primarily maintained for egg production. We did not include males in the current experiment, and therefore, the potential influence of sex on fear-related behaviors was not assessed in this study. However, we fully agree that sex differences may play a role in fear responses and behavioral traits. This is an important factor that deserves further investigation, and we appreciate your suggestion. We will consider incorporating this aspect in our future research.

Comments 2: In this study, lines 133-138 describe the use of the left hemisphere of the chicken brain for histological observation, while hippocampal tissues were used for transcriptome sequencing and PCR assays. However, lines 140-147 mention eosin staining being performed on hippocampal tissues. Therefore, was the histological observation conducted on the hippocampal tissues or the left brain? Please clarify.

Response 2: Thank you for pointing out this potential confusion. We apologize for the lack of clarity in the manuscript. To clarify: hematoxylin and eosin (H&E) staining was performed on sections from the left hemisphere of the chicken brain. During histological observation, hippocampal regions were identified under a light microscope at 400× magnification for morphological analysis. We have revised the relevant sentences in the Materials and Methods section to clearly state that histological observations were specifically conducted on the hippocampal regions within the left brain.

We change the original sentence to :“The fixed left hemispheres of the chicken brains were embedded in paraffin and sectioned at 5 μm thickness. Sections were immersed in various concentration gradients of xylene and anhydrous ethanol, followed by staining with hematoxylin and eosin. After transparency treatment, they were embedded in neutral gum and observed and photographed using an orthogonal white light photomicrographic microscope (Nikon Eclipse Ci-L, Nikon, Tokyo, Japan). During microscopic examination, the hippocampal regions were identified within the stained sections of the left hemisphere based on anatomical landmarks. The histomorphometric structure of the hippocampus was then evaluated under high magnification (40×) using CaseViewer 2.4 (3DHISTECH Ltd., Budapest, Hungary). ”

Comments 3: In lines 396-398, the authors present their hypothesis. If this hypothesis is indeed original and based on their own research findings, why is a reference cited? If a citation is necessary, does this imply that the hypothesis was originally proposed by others and you are merely validating it? Please clarify this discrepancy.

Response 3: Thank you for your question. The hypothesis proposed in lines 396-398 was developed based on insights from our own research results, and was also inspired and supported by previous studies (references 20 and 21). However, we acknowledge that there was an error in the placement of reference 21 in the original manuscript, which may have caused you confusion about the source of the hypothesis. We have now corrected the location of the reference to accurately reflect that the hypothesis is an original proposition based on our experimental results, but with reference to previous literature. Thank you for your attention to detail and have modified the text accordingly to make it clearer.

Round 2

Reviewer 2 Report

Comments and Suggestions for Authors

The authors have responded to the comments I provided in the first revision.

            Comments on the Quality of English Language

Reviewer 3 Report

Comments and Suggestions for Authors

Although some of my concerns are still present, addressing them without redesigning the experiment would not be possible. At the same time, authors in their response provided adequate reasoning for their decisions. The manuscript has been considerably improved.